# Complex Microbial Infection of Urachal Remnant: A Case Report

**DOI:** 10.3390/reports8040255

**Published:** 2025-12-03

**Authors:** Koji Tajima, Tsuneaki Kenzaka, Ayaka Arimoto, Shota Nokubo, Hisanobu Deguchi

**Affiliations:** 1Division of Internal Medicine, Kokuho Nokami Kosei Sogo Hospital, 198 Shobata, Kimino-cho, Kaiso-gun, Wakayama 640-1141, Japan; a.alucky140rs8@gmail.com (A.A.); s-nokubo@wakayama-med.ac.jp (S.N.); deguchi_h@nokami-hospital.jp (H.D.); 2Division of Surgery, Kokuho Nokami Kosei Sogo Hospital, 198 Shobata, Kimino-cho, Kaiso-gun, Wakayama 640-1141, Japan; 3Division of Community Medicine and Career Development, Kobe University Graduate School of Medicine, 2-1-5, Arata-cho, Hyogo-ku, Kobe 652-0032, Japan; smile.kenzaka@jichi.ac.jp

**Keywords:** urachal remnant, *Corynebacterium* sp., Bacteroides ovatus, *Anaerococcus vaginalis*, *Bacteroides uniformis*, *Peptostreptococcus stomatis*, anaerobic bacteria infection

## Abstract

**Background and Clinical Significance:** We present a rare case of an infected urachal remnant involving four microorganisms, including anaerobic bacteria. **Case Presentation:** A 23-year-old man presented with abdominal pain around the umbilicus, diarrhea and discharge. Laboratory findings and imaging led to a diagnosis of an infected urachal remnant. He was treated with broad-spectrum antibiotic therapy, and the abscess in the urachal remnant was drained. In cultures from the purulent urachal remnant, *Bacteroides ovatus*, *Anaerococcus vaginalis*, *Bacteroides uniformis*, and *Peptostreptococcus stomatis* were detected. After 2 months, the urachal remnant infection had not relapsed. **Conclusions:** This report presents the first documented case of an infected urachal remnant in which four anaerobic microorganisms were identified. In patients with fever, abdominal pain, and discharge from the umbilicus, physicians should consider the possibility of an infected urachal remnant in their differential diagnosis. Treatment should include appropriate antibiotic therapy to cover anaerobic organisms, and in cases where the clinical course does not improve, drainage of the urachal remnant may be necessary.

## 1. Introduction and Clinical Significance

The allantois is an embryological structure that connects the placenta and cloaca. The allantois regresses before delivery and becomes the urachus, a ligament extending between the bladder and umbilicus [1]. Incomplete closure of the urachus can result in infection or carcinogenesis. The reported prevalence of urachal remnants varies widely, ranging from 1:100 to 1:150,000, depending on the report; however, a higher prevalence is commonly reported in male individuals than in female individuals [2]. Antibiotic agents and drainage are often used for treatment. After the infection is controlled, radical surgery is recommended. An infected urachal remnant is mainly caused by a combination of microorganisms; therefore, a broad-spectrum antibiotic therapy should be selected [3]. However, to the best of our knowledge, no reports mention the number of microorganisms detected in a single case.

Here, we sought to present a rare case in which the patient’s symptoms worsened despite antibiotic treatment. The aim of this report is to provide evidence that the infected urachal remnant is a cause of fever, abdominal pain, and discharge from the umbilicus.

## 2. Case Presentation

A 23-year-old man presented with discharge of pus from his umbilicus. Three days before presentation, he consulted a primary care physician because he had experienced diarrhea and abdominal pain around his umbilicus. He was diagnosed with infectious colitis and prescribed oral clarithromycin (400 mg per day). On the day he visited our hospital, he noticed pus discharge from his umbilicus and revisited his primary care physician, who then referred the patient to our hospital. The patient’s medical history was unremarkable.

His presenting vital signs were a blood pressure of 144/88 mmHg, a heart rate of 85 beats/min, a body temperature of 36.8 °C, a respiratory rate of 16 breaths/min, and an oxygen saturation of 98% on room air. The abdomen was flat and non-distended; however, mild tenderness to palpation and tapping pain around the umbilicus were observed. Carnett’s sign was positive, suggesting an extraperitoneal lesion. Bowel sounds remained unchanged. A small amount of pus was observed in the umbilicus, which did not increase upon pressing around the umbilicus (Figure 1).

Blood test results revealed the following: white blood cell count, 10,500/μL; hemoglobin level, 14.7 g/dL; platelet count, 23.7 × 10^4^/μL; blood urea nitrogen level, 13.7 mg/dL; creatinine level, 0.91 mg/dL; and C-reactive protein level, 0.74 mg/dL (Table 1). These results indicated mildly elevated inflammation marker levels; however, they did not indicate hepatitis, cholecystitis, or pancreatitis. Abdominal ultrasound sonography (A-US) revealed a low-echoic lesion of approximately 10 mm just below the umbilicus. Abdominal computed tomography (A-CT) revealed a soft tissue image below the umbilicus (Figure 2).

The patient was diagnosed with an infected urachal remnant and prescribed amoxicillin/clavulanic acid (1500 mg/375 mg per day). Subsequently, he was instructed to wash his umbilicus carefully. Three days after his visit, he returned to the hospital, reporting fever and worsening abdominal pain. His body temperature was 37 °C. Upon physical examination, redness around his umbilicus was observed. The tenderness and tapping pain were worse, and the painful area had expanded since the previous visit. The amount of pus from his umbilicus had also increased (Figure 3).

The A-US and A-CT findings suggested the enlargement of the abscess below the umbilicus. Enhanced A-CT revealed multilocular abscesses (Figure 4 and Figure 5). The abscess was drained with the patient under sedation, with local anesthesia administered as follows: intravenous midazolam (3 mg), intravenous pentazocine (15 mg), and subcutaneous injection of lidocaine (8 mg). When the base of the umbilicus was opened with mosquito forceps, a large amount of purulent discharge was observed (Figure 6). The umbilicus was carefully washed, and a drainage tube was inserted at its base. The Gram stain of the pus showed a multi-bacterial pattern including Gram-positive cocci, Gram-positive rods, and Gram-negative rods (Figure 7). He was hospitalized at our hospital, and the oral antibiotic was changed to intravenous ampicillin/sulbactam (3 g every 8 h).

On day 1 after admission, body temperature improved, and the inflammation marker levels decreased. On day 5, the drainage tube was removed. On day 10, the antibiotic was changed to cefalexin (750 mg per day) and metronidazole (1500 mg per day), and the patient was discharged.

Two days after discharge, he visited our outpatient clinic, and his laboratory test results revealed mild liver enzyme elevation. Therefore, the amount of metronidazole administered was decreased by half (750 mg per day). Fifteen days after discharge, the course of antibiotics was completed, and he was referred to another hospital for a radical operation. After a period of 2 months, the urachal remnant infection did not recur.

At the initial visit, a swab sample was obtained from the purulent discharge at the umbilicus for bacterial culture. The culture yielded *Bacteroides ovatus* and *Anaerococcus vaginalis*, with semiquantitative assessments of 1+ and 2+, respectively.

The semi-quantitative bacterial grading (1+, 2+, etc.) was performed according to the microbiology laboratory’s standard operating procedures to ensure consistent estimation of bacterial load. The criteria were as follows: 1+: A few colonies are observed across the entire field of view (estimated 10^3^ cfu/mL). 2+: One to a few colonies are observed in every field of view (estimated 10^4^ cfu/mL). 3+: Several tens or more colonies are observed in every field of view (estimated 10^5^ cfu/mL or more).

The isolates were definitively identified using matrix-assisted laser desorption/ionization time-of-flight mass spectrometry (MALDI Biotyper sirius one, Bruker Daltonics, Bremen, Germany).

Subsequently, during the drainage procedure, a second specimen was obtained by directly aspirating the discharged pus into a syringe. Culture of this specimen yielded *Bacteroides uniformis* and *Peptostreptococcus stomatis*, both graded 1+ on semiquantitative assessment.

The isolates were cultured aerobically on blood agar and chocolate agar and anaerobically on Columbia blood agar under anaerobic conditions. Antimicrobial susceptibility testing was performed using the disk diffusion method, and results were interpreted according to the Clinical and Laboratory Standards Institute guidelines. This globally accepted system establishes specific interpretive criteria (breakpoints) for microorganisms and antimicrobial agents. Results are primarily categorized into three clinical outcomes, based on the measured inhibition zone diameters (in millimeters): susceptible (S), intermediate (I), and resistant (R). These categories guide clinicians in selecting effective treatment based on standardized, reproducible testing methods. The isolates exhibited resistance to macrolide antibiotics, and additional susceptibility results are summarized in Table 2.

## 3. Discussion

In this report, we present a case of an infected urachal remnant. The patient was initially misdiagnosed with infectious colitis due to the presentation of abdominal pain and diarrhea and was prescribed an antibiotic agent. However, further investigation revealed the involvement of various bacterial species, including anaerobic bacteria, explaining the ineffectiveness of the antibiotic therapy. Infected urachal remnants due to anaerobic bacteria are common; however, a case involving four microorganism isolates in an infected urachal remnant with purulence is extremely rare. Uehara et al. reported different cases of infected urachal remnants, where each was caused by a combination of microorganisms (82%); however, the number of species involved in each case was not mentioned [3]. Additionally, other previous reports showed single-organism involvement or did not focus on the number of involved organisms [4,5,6,7,8]. In some cases, cultures showed no growth [7,8,9,10].

The isolation of these four anaerobic species in the present case strongly suggests that the infection originated primarily from the adjacent gastrointestinal and/or genitourinary microbiota (commensal flora). Specifically, these anaerobes, including *Bacteroides* sp., *Anaerococcus* sp., and *Peptostreptococcus* sp., are well-recognized constituents of the commensal flora of these tracts [11,12,13]. Infection of urachal remnants is observed to occur through the blood, lymph, or directly from the bladder [14]. In a similar case report of an infected urachal remnant, the same pathogens were identified from blood and urine cultures [4]. Newman et al. reported that the common pathogen of infected urachal remnants is *Staphylococcus aureus,* which is a part of the normal skin flora [5]. However, from a single-center retrospective study in Japan, the most isolated bacteria were *Prevotella* sp. (9%), *Bacteroides* sp. (8%), *Staphylococcus* sp. (6%), *Streptococcus* sp. (6%), and *Corynebacterium* sp. (6%) [3]. The study by Newman et al. involved a relatively small series involving children aged 8 months to 9 years, with the samples from only five of six patients undergoing culture analysis; of these, four were positive and three yielded *S. aureus*. In contrast, the study by Uehara et al. was comparatively larger, analyzing 93 culture samples and including a population in which 36% were aged under 20 years, indicating a relatively higher proportion of adults. Therefore, the differences in etiology between the two reports may be largely influenced by both the research scale and age distribution of the patient populations.

In the present case, the patient was prescribed clarithromycin for infectious colitis due to his diarrheal symptoms. However, three of the four isolated anaerobic bacteria were resistant to macrolide, and the remaining isolate showed intermediate susceptibility. Therefore, a treatment regimen that targets the common and causative pathogens, including the anaerobic microorganisms mentioned earlier, is recommended. In the cases reported by Tazi et al., the antibiotic regimens used were amoxicillin/clavulanic acid + gentamycin and ceftriaxone + gentamycin [6]. Tawk et al. selected a combination antibiotic regimen of penicillin and aminoglycoside to target both Gram-positive and Gram-negative organisms [9]. Similarly, Hassan et al. reported a case in which intravenous ceftriaxone was administered to a child [10]. Kashiwagi et al. initially used ceftriaxone and later de-escalated to ampicillin after culture results were confirmed [4]. Pediatric cases of infected urachal remnant may involve more aerobic bacteria, such as *Escherichia coli*, *Staphylococcal* sp., and *Streptococcus* sp., or anaerobic bacteria, such as *Peptostreptococcus* sp., which are susceptible to ceftriaxone [7]. Broad-spectrum antibiotic therapy, such as ampicillin/sulbactam or amoxicillin/clavulanic acid, or combined therapy, such as metronidazole with cefalexin or amoxicillin, appears appropriate. When initial drainage of the abscess in an infected urachal remnant is performed, coverage for anaerobic bacteria may not always be necessary [15].

Diagnosing infected urachal remnants is difficult. One of the reasons for this difficulty is the rarity of infected urachal remnants in primary and emergency care. Additionally, the symptoms of infected urachal remnants often mimic those of other acute abdominal diseases; the most common symptom is umbilical pain or discharge [2,7]. However, another study reports that the most common symptom is abdominal pain [8]. If there is little discharge from the umbilicus or if a physician examines the abdomen without specifically inspecting the umbilicus, an infected urachal remnant may be overlooked. Moreover, in cases similar to that presented in this report, when the discharge from the umbilicus is not observed or noticed at the time of initial visit and the patient complains only of abdominal pain, diagnosis is challenging. When the patient in the current case presented at our hospital, Carnett’s sign was positive upon physical examination. Carnett’s sign may be useful in distinguishing extraperitoneal diseases, such as infected urachal remnants, from intraperitoneal diseases [16].

In the present case, the culture results may have been affected by previous antibiotic therapy. In other words, the detection of only macrolide-resistant (intermediate) bacteria could be attributed to the disappearance of macrolide-susceptible bacteria following clarithromycin administration. This limitation must be considered when interpreting our culture findings. In the future, more detailed studies on the presence of pathogens in infected urachal remnants are needed to determine the suitable antibiotic therapy.

To the best of our knowledge, this study reports the first known case of an infected urachal remnant from which five microorganisms, including anaerobic bacteria, were isolated. In patients with fever, abdominal pain, or discharge from the umbilicus, physicians must consider a differential diagnosis of an infected urachal remnant.

## 4. Conclusions

Antibiotic therapy targeting anaerobic organisms should be initiated for infected urachal remnants. In cases where the clinical course does not show improvement, drainage of the urachal remnant may be necessary.

## Figures and Tables

**Figure 1 reports-08-00255-f001:**
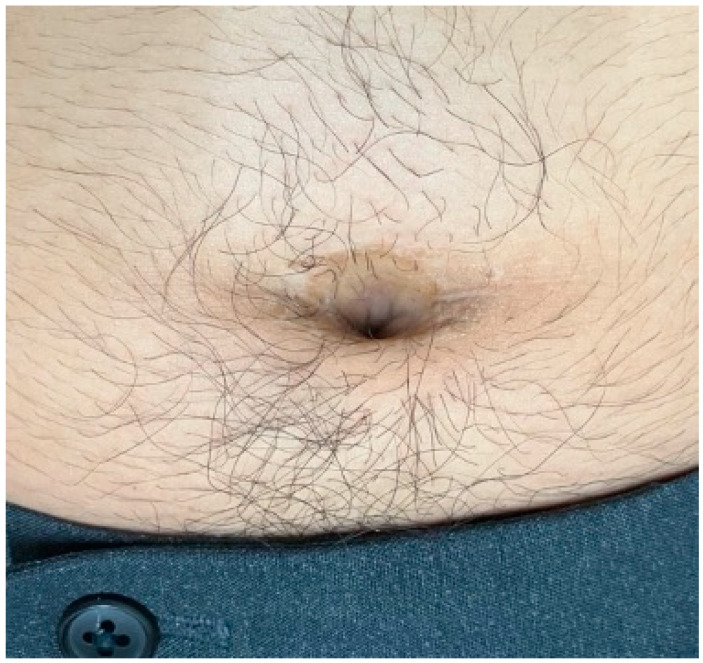
Gross findings of the pooled pus in the umbilicus at the initial visit.

**Figure 2 reports-08-00255-f002:**
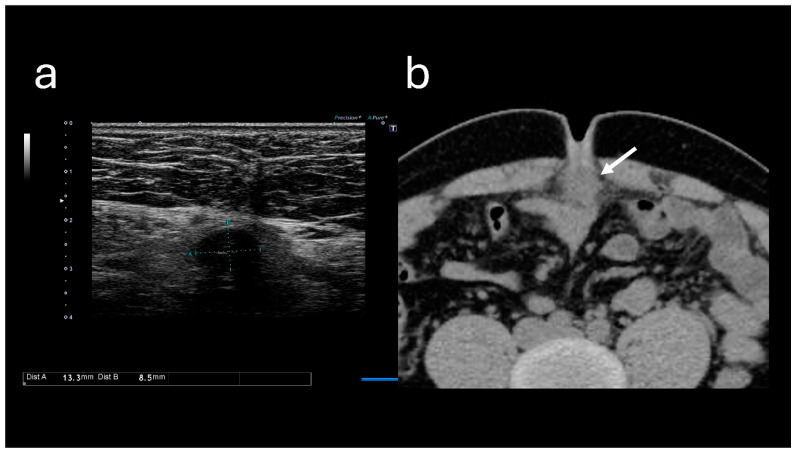
Images obtained at the time of the initial visit. (**a**) Abdominal ultrasound sonography reveals a low-echoic lesion of approximately 10 mm just below the umbilicus. (**b**) Abdominal computed tomography reveals a soft tissue mass located below the umbilicus (arrow).

**Figure 3 reports-08-00255-f003:**
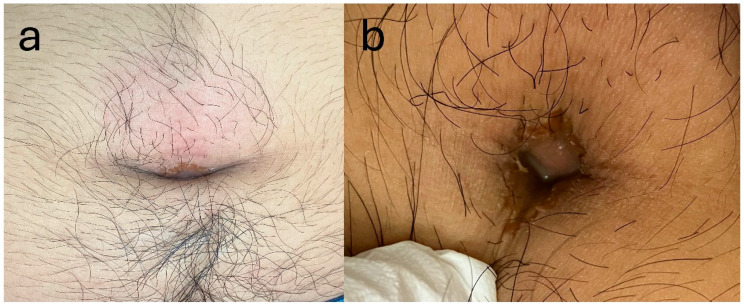
Gross findings of the umbilicus at the second visit. (**a**) Marked redness around the umbilicus. (**b**) Increased pus compared with that at the initial visit.

**Figure 4 reports-08-00255-f004:**
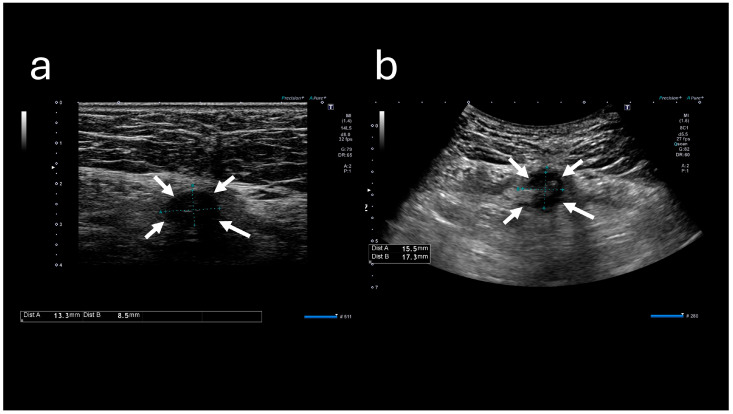
Abdominal ultrasound sonography images. (**a**) Image obtained at the time of initial visit. (**b**) Image obtained at the time of the second visit. The low-echoic lesion (indicated by arrows) appears enlarged.

**Figure 5 reports-08-00255-f005:**
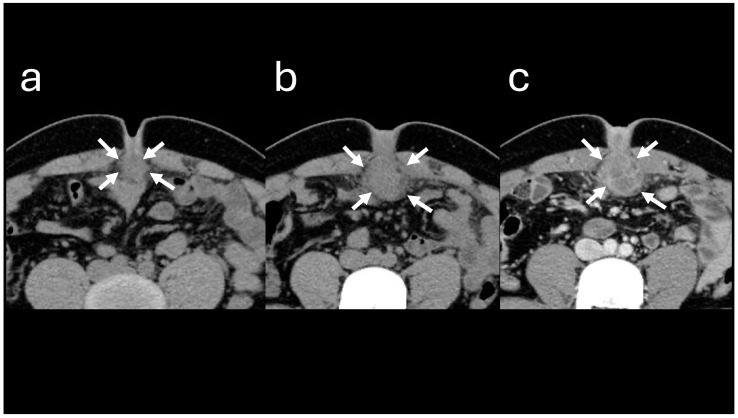
Abdominal computed tomography images. (**a**) Image obtained at the time of initial visit. (**b**) Image obtained at the time of the second visit. The soft tissue lesion below the umbilicus (indicated by arrows) appears enlarged. (**c**) Contrast-enhanced image demonstrates a multicapsulated soft tissue lesion.

**Figure 6 reports-08-00255-f006:**
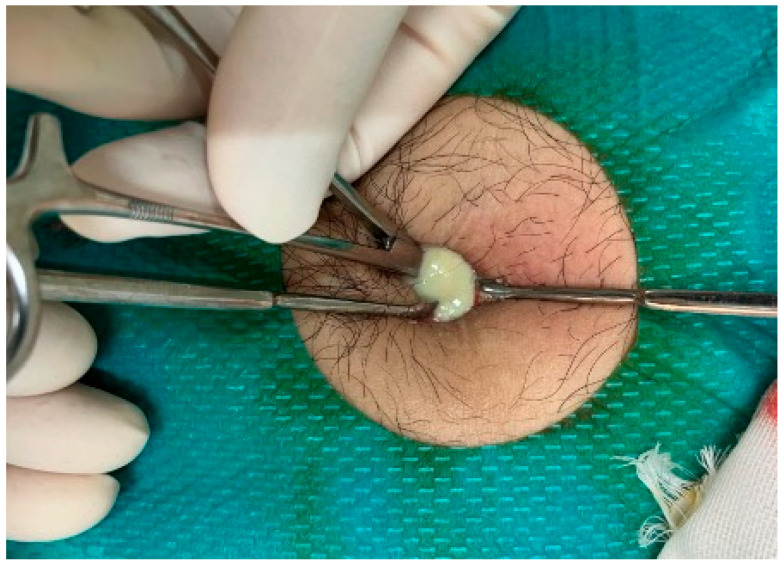
Image of the drainage of an abscess below the umbilicus. A large amount of purulent discharge was released immediately upon opening the abscess with mosquito forceps.

**Figure 7 reports-08-00255-f007:**
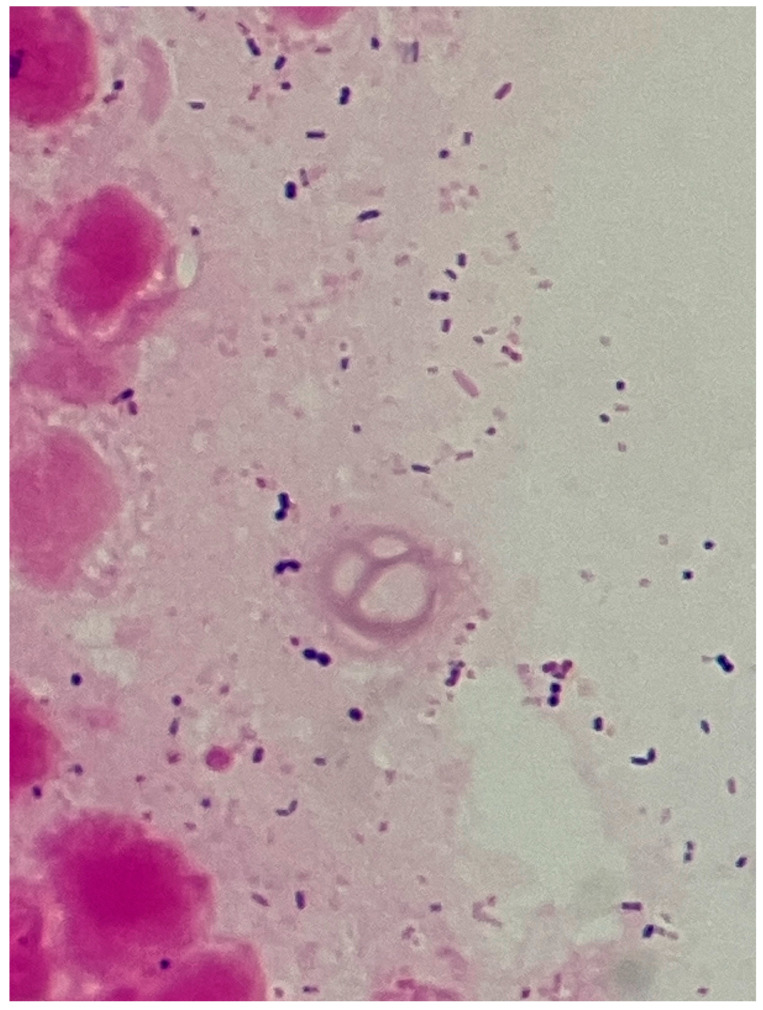
Gram-stain image (magnification: ×5000). A multi-bacterial pattern is observed, with Gram-positive cocci, Gram-positive rods, and Gram-negative rods.

**Table 1 reports-08-00255-t001:** Laboratory data at the initial visit.

Parameter	Recorded Value	Standard Value
White blood cell count	10,500/μL	3800–9500/μL
Neutrophils	67.7%	30–70%
Red blood cell count	458 × 10^4^/μL	410–530 × 10^4^/μL
Hemoglobin level	14.7 g/dL	13.0–18.0 g/dL
Hematocrit	39.6%	38–50%
Platelet count	23.7 × 10^4^/μL	12–35 × 10^4^/μL
Total protein level	7.8 g/dL	6.5–8.3 g/dL
Aspartate aminotransferase level	20 U/L	13–33 U/L
Alanine aminotransferase level	50 U/L	6–35 U/L
Lactate dehydrogenase level	202 U/L	124–222 U/L
γ-glutamyl transferase level	24 U/L	11–64 U/L
Total bilirubin level	1.1 mg/dL	0.2–1.2 mg/dL
Amylase level	36 U/L	40–150 U/L
Creatinine phosphokinase level	124 U/L	61–255 U/L
Glucose level	99 mg/dL	60–110 mg/dL
Blood urea nitrogen level	13.7 mg/dL	8.0–22.0 mg/dL
Creatinine level	0.91 mg/dL	0.6–1.2 mg/dL
Sodium level	143 mEq/L	136–146 mEq/L
Potassium level	4.6 mEq/L	3.5–5.1 mEq/L
C-reactive protein level	0.74 mg/dL	<0.30 mg/dL

**Table 2 reports-08-00255-t002:** Antibiotic susceptibility of bacteria in this case.

	*Bacteroides ovatus*	*Anaerococcus vaginalis*	*Bacteroides uniformis*	*Peptostreptococcus stomatis*
ABPC	S	S	R	S
A/S	S	S	S	S
PIPC	S	S	S	S
P/T	S	S	S	S
CEZ	I	S	I	S
CTM	I	S	R	S
C/S	S	S	S	S
CAZ	R	S	R	S
CTRX	I	S	R	S
CFPM	R	S	R	S
CMZ	I	S	S	S
MEPM	S	S	S	S
AZM	R	I	R	R
CLDM	R	S	R	S
MINO	I	S	I	S
VCM	R	S	R	S
LVFX	S	S	R	S

ABPC: ampicillin, A/S: ampicillin/sulbactam, PIPC: piperacillin, P/T: piperacillin/tazobactam, CEZ: cefazolin, CTM: cefotiam, C/S: cefoperazone/sulbactam, CAZ: ceftazidime, CTRX: ceftriaxone, CFPM: cefepime, CMZ: cefmetazole, MEPM: meropenem, AZM: azithromycin, CLDM: clindamycin, MINO: minocycline, VCM: vancomycin, LVFX: levofloxacin, S: susceptible, I: intermediate, R: resistant.

## Data Availability

The original contributions presented in this study are included in the article. Further inquiries can be directed to the corresponding author.

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
