# Peer review of "Complex Microbial Infection of Urachal Remnant: A Case Report"

_reports, 2025, doi:10.3390/reports8040255_

Round 1

Reviewer 1 Report

Comments and Suggestions for Authors

This case report by Tajima et al. describes a case urachal remnant infection in a 23-year-old man. The report is well written and detailed. However, the significance of the report is based upon two factors, the case itself with medical diagnosis (excellent) and the bacterial etiology of what the authors report is a complex infection. No data is provided to support the later factor.

  1. The authors must provide laboratory data to support the identification of five bacterial species, four of which are anaerobes. How were these identified and what system was used to generate a susceptibility profile? Were any MIC’s performed? Why was Corynebacterium not identified to species? Yes, many corynebacteria are skin contaminants but others, such as C. striatum as one example, are not. The Gram stain in figure 7 seems to suggest it could be significant.
  2. How do the authors know which of these bacteria were causing infection? Was any semi-quantitative count performed? What media were used for their isolation? How do the authors know Corynebacterium spp. was a contaminant?
  3. Delete figure 8. The text adequately addresses the subject.
  4. I am not sure I see the medical significance of five bacteria being recovered (line 39-40) from this patient vs a smaller number of species. Wouldn’t you cover with antimicrobial agents for multiple groups when two or more species are isolated simultaneously? With anaerobe recovery, you are never sure you have isolated all the infecting agents.

Author Response

Dear Reviewer 1

We are grateful for the thorough and constructive feedback provided on our manuscript. We have addressed all comments and suggestions meticulously. We believe the manuscript has been significantly improved in terms of clarity, accuracy, and scientific rigor, particularly concerning the detailed microbiological analysis.

Comment 1: The authors must provide laboratory data to support the identification of five bacterial species, four of which are anaerobes. How were these identified and what system was used to generate a susceptibility profile? Were any MIC’s performed? Why was Corynebacterium not identified to species? Yes, many corynebacteria are skin contaminants but others, such as C. striatum as one example, are not. The Gram stain in figure 7 seems to suggest it could be significant.

Response 1: We acknowledge the critical importance of providing detailed laboratory methods to support our findings, especially concerning the significance of Corynebacterium sp. We have clarified the identification and susceptibility testing procedures accordingly.

Anaerobic Bacteria Identification: The four anaerobic species were identified using the Mass Spectrometry system (MALDI Biotyper sirius one; Bruker Daltonics).

Antimicrobial Susceptibility Testing (AST): The susceptibility profiles were generated using the disk diffusion method. We confirm that Minimum Inhibitory Concentrations (MICs) were not determined, as the disk diffusion method provides category interpretations (Susceptible, Intermediate, Resistant) rather than specific MIC values.

Corynebacterium sp. Identification: The identification of Corynebacterium sp. was based solely on its Gram-staining characteristics (Gram-positive coryneform rods) and colony morphology on the culture media. Although species-level identification would have enhanced the accuracy of the report, molecular testing was not performed due to the additional cost, as the patient was not willing to bear the expense after consultation.

Significance of Corynebacterium: We recognize the potential significance of this organism, as highlighted by the Gram stain (Figure 7). Our primary objective is to emphasize to the reader that infected urachal remnants are highly polymicrobial. We assert that the recovery of Corynebacterium sp. alongside multiple strict anaerobes—regardless of the definitive species identity—adequately serves this purpose, as it supports the possibility of a complex infection involving both skin and bowel flora.

We have added details of the microbiological examinations to the main text (Lines 122–140).

Comment 2: How do the authors know which of these bacteria were causing infection? Was any semi-quantitative count performed? What media were used for their isolation? How do the authors know Corynebacterium spp. was a contaminant?

Response2 : We thank the reviewer for raising these fundamental questions regarding the pathogenicity of the isolated organisms.

We agree that there are no absolute means to definitively determine whether every single identified species was actively causing the infection. However, our clinical and microbiological interpretation is based on the following:

  • Semi-Quantitative Counts: The semi-quantitative test results did not show a difference so extreme that any single isolate could be readily dismissed as a contaminant.
  • Polymicrobial Pathogenicity: Infected urachal remnants are well-documented to be caused by polymicrobial infections involving both skin and bowel flora. Given the potential pathogenicity of the isolated species (e.g., strict anaerobes) and nature of the closed-space infection, we determined it was clinically appropriate to treat all isolated organisms as relevant pathogens contributing to the infection.
  • Media Used for Isolation: For aerobic bacteria, isolates were cultured using blood agar medium and chocolate agar medium. For anaerobic bacteria, Anaero Columbia blood agar medium was used.

We have included the results of the semi-quantitative test and the specific types of culture media used in the revised manuscript to enhance transparency and support our interpretation.

Comment 3: Delete figure 8. The text adequately addresses the subject.

Response 3: We agree with the reviewer that the content of Figure 8 is already well covered within the text of the manuscript. As requested, we have deleted Figure 8 from the manuscript.

Comment 4: I am not sure I see the medical significance of five bacteria being recovered (line 39-40) from this patient vs a smaller number of species. Wouldn’t you cover with antimicrobial agents for multiple groups when two or more species are isolated simultaneously? With anaerobe recovery, you are never sure you have isolated all the infecting agents.

Response 4: We appreciate the reviewer's insightful clinical observation. We fully agree that the distinction between four and five recovered species may not significantly alter the immediate empirical antimicrobial management, as the isolation of two or more organisms, especially anaerobes, already mandates a broad-spectrum regimen covering multiple bacterial groups.

However, the clinical value and significance of this case report lie in providing specific microbiological evidence of the complex and highly polymicrobial nature of infected urachal remnants. While previous reports have acknowledged the involvement of multiple organisms, they often lack the detailed species count and identification that underscore the true complexity of this infection.

Therefore, the detailed data showing five distinct species serve to strongly reinforce the following crucial educational message to the reader:

Infected urachal remnants are frequently highly polymicrobial, a finding that should immediately direct clinicians toward the selection of broad-spectrum empirical antimicrobial agents, with mandatory coverage for anaerobes, prior to receiving culture results.

Ultimately, we would like to express our sincere gratitude to the editors and reviewers for their positive and constructive criticism. The manuscript has vastly benefited from your valuable and insightful comments and suggestions. We look forward to hearing from you and would be happy to address any further concerns, if required. We hope this further pushes the manuscript closer to publication in your esteemed journal.

All authors have read and approved the revised manuscript. 

Sincerely,

Koji Tajima, MD

Reviewer 2 Report

Comments and Suggestions for Authors

The paper by Tajima et al details the case of a patient whose urachal remnant was infected.  The study is well written, has good quality images and detailed patient test data.  The paper covered well the presentation of patient symptoms and the clinical therapy and overall outcome.  Given urachal remnant infections are so rare, this is a very useful reference study for physicians and microbiologists generally.

Author Response

Dear Reviewer 2

We are grateful for the thorough and constructive feedback provided on our manuscript. We have addressed all comments and suggestions meticulously. We believe the manuscript has been significantly improved in terms of clarity, accuracy, and scientific rigor, particularly concerning the detailed microbiological analysis.

Comment: The paper by Tajima et al details the case of a patient whose urachal remnant was infected.  The study is well written, has good quality images and detailed patient test data.  The paper covered well the presentation of patient symptoms and the clinical therapy and overall outcome.  Given urachal remnant infections are so rare, this is a very useful reference study for physicians and microbiologists generally.

Response: We deeply appreciate the reviewer's extremely positive comments on the quality of our manuscript, images, and detailed patient data.

We are particularly pleased that the reviewer recognizes the usefulness of this study as a reference for physicians and microbiologists, given the rarity of infected urachal remnant cases. Our primary goal was to provide a comprehensive and detailed account of the presentation, clinical therapy, and microbiological findings to assist others facing this challenging diagnosis.

Thank you for your valuable time and encouraging feedback.

Sincerely,

Koji Tajima, MD

Reviewer 3 Report

Comments and Suggestions for Authors

The case report is really interesting and deserves to be published since urachal remnant infections are so rare. However, there are some issues that needs to be addressed.

  1. Line 44 – replace types to species
  2. Line 45 – the sentence” we recommend that antibiotic therapy should cover anaerobic organism” is random in the text. The information is accurate however it needs to be rephrased and perhaps moved somewhere else and also add a citation to that information since the recommendation is not based on this study.
  3. Line 61 – replace purulence with pus
  4. Table 1 – I encourage you to highlight the abnormal values so they are easier to spot
  5. Figure 3 – explanation for b needs to be rephrase and perhaps change purulence with pus.
  6. Figure 4 and 5 – please add arrows on the picture where the area of interest is
  7. My main concern with this paper is that authors failed to establish a correlation between the microorganism that were found in the pus and the infections since most of them are part of the normal local flora and they were recovered from two different samples. Please add a complete microbiologic diagnosis including the specifics about how the samples were taken, how were bacteria identified and how was performed the antimicrobial susceptibility testing.
  8. Table 2 – include the species in the table instead of numbers and explanations bellow
  9. Please state according to which guideline the antimicrobial susceptibility testing was reported
  10. Line 155-157 – please rephrase the entire sentence
  11. Line 165 – Does not make sense to me – please rephrase
  12. Citations need to be uniformized according to the journals guideline

Author Response

Dear Reviewer 3

We are grateful for the thorough and constructive feedback provided on our manuscript. We have addressed all comments and suggestions meticulously. We believe the manuscript has been significantly improved in terms of clarity, accuracy, and scientific rigor, particularly concerning the detailed microbiological analysis.

Comment 1: Line 44 – replace types to species

Response 1: Thank you for your meticulous review. We agree that "species" is the correct and more specific term in this context. We have replaced "types" with "species" as suggested.

Comment 2: Line 45 – the sentence” we recommend that antibiotic therapy should cover anaerobic organism” is random in the text. The information is accurate however it needs to be rephrased and perhaps moved somewhere else and also add a citation to that information since the recommendation is not based on this study.

Response 2: We agree with the reviewer that the original sentence, which provided a general therapeutic recommendation, was misplaced and lacked context at the end of the Introduction section. We have deleted the sentence ("We recommend that antibiotic therapy should cover anaerobic organisms. Moreover, when the clinical course does not improve under appropriate antibiotic therapy, drainage of the urachal remnant may be necessary.") from the Introduction accordingly.

Comment 3: Line 61 – replace purulence with pus

Response 3: Thank you for the suggestion. We agree that "pus" is the appropriate noun for the substance being described in this context, as opposed to "purulence" (which describes the state or quality of being purulent). We have replaced "purulence" with "pus" as requested.

Comment 4: Table 1 – I encourage you to highlight the abnormal values so they are easier to spot

Response 4: Thank you for this constructive suggestion. We agree that highlighting the abnormal values will significantly improve the readability and utility of Table 1. We have revised Table 1 to visually emphasize the abnormal values:

  • Values exceeding the normal upper limit are highlighted in red.
  • Values falling below the normal lower limit are highlighted in blue.

Comment 5: Figure 3 – explanation for b needs to be rephrase and perhaps change purulence with pus.

Response 5: Thank you for the suggestion regarding the explanation for Figure 3b. We agree that the description was unclear and that "pus" is the more accurate term for the visual content than "purulence." We have rephrased the explanation for Figure 3b for clarity and have replaced "purulence" with "pus" as requested.

Comment 6: Figure 4 and 5 – please add arrows on the picture where the area of interest is

Response 6: Thank you for this constructive suggestion. We agree that visual clarity is essential for interpreting the images. We have revised both Figures 4 and 5 by adding arrows to clearly indicate the area of interest (the lesion) in each picture, thereby improving the visualization for the reader.

Comment 7: My main concern with this paper is that authors failed to establish a correlation between the microorganism that were found in the pus and the infections since most of them are part of the normal local flora and they were recovered from two different samples. Please add a complete microbiologic diagnosis including the specifics about how the samples were taken, how were bacteria identified and how was performed the antimicrobial susceptibility testing.

Response 7: We thank the reviewer for addressing this critical concern regarding the pathogenicity of the recovered microorganisms.

We agree that it is absolutely difficult to definitively determine whether every single identified bacterium was a true pathogen or merely a contaminant. However, given the clinical context and microbiological data, we managed the case under the assumption that all isolated organisms were contributing to the infection, based on the following reasons:

  • Infected urachal remnants are characteristically polymicrobial infections.
  • The infection occurred within a closed-space environment.
  • The isolated organisms, particularly the strict anaerobes, possess potential pathogenicity.

To provide a microbiological diagnosis, we have added the following specifics regarding specimen collection and testing methods to the revised manuscript:

Specimen Collection Details

The specimens were collected on two separate occasions using different techniques:

First Sample: Pus accumulated at the umbilicus was collected using a swab.

Second Sample: Pus from the abscess cavity was aspirated using a sterile syringe during the drainage procedure.

Identification and Susceptibility Testing

Corynebacterium sp.: Identification was based solely on Gram-staining characteristics (coryneform rods) and colony morphology. Mass spectrometry was not utilized for this isolate due to cost constraints.

Anaerobic Bacteria: The anaerobic species were identified using mass spectrometry, specifically the MALDI Biotyper Sirius One system (Bruker Daltonics).

Antimicrobial Susceptibility Testing (AST): AST was performed using the disk diffusion method. The results were interpreted according to the Clinical and Laboratory Standards Institute (CLSI) guidelines.

We have added details of the microbiological examinations to the main text (Lines 118–136).

Comment 8: Table 2 – include the species in the table instead of numbers and explanations bellow

Response 8: Thank you for this constructive suggestion regarding the presentation of Table 2. We agree that listing the bacterial species directly in the table, rather than using codes and explanations below, significantly enhances clarity. We have revised Table 2 to include the bacterial species names directly within the table, replacing the previous numbered system and its corresponding explanatory footnote.

Comment 9: Please state according to which guideline the antimicrobial susceptibility testing was reported

Response 9: Thank you for this important inquiry. We recognize that stating the reference guideline is essential for the interpretation and reproducibility of our antimicrobial susceptibility data. We confirm that the interpretation of the antimicrobial susceptibility testing was performed according to the guidelines established by the Clinical and Laboratory Standards Institute (CLSI). We have included this specific detail in the revised manuscript.

Comment 10: Line 155-157 – please rephrase the entire sentence

Response 10: Thank you for this essential comment. We agree that the original sentence, which merely stated that the isolated bacteria were "normal flora," failed to establish the clinical significance of the findings. We have rephrased the sentence (now lines 155–158) to clearly state the implication of these findings for the infection's origin:

The isolation of these five species strongly suggests that the infection originated from the adjacent skin and/or bowel microbiota (commensal flora). Specifically, Corynebacterium sp. isrecognized as an indigenous skin microbe, while the strict anaerobes (Bacteroides sp., Anaerococcus sp., and Peptostreptococcus sp.) are typical constituents of the gastrointestinal and genitourinary tracts.

This revision establishes the source of the infection and replaces the outdated "normal flora" term with the more appropriate "microbiota (commensal flora)."

Comment 11: Line 165 – Does not make sense to me – please rephrase

Response 11: We apologize for the lack of clarity in the original paragraph. We recognize that the previous text was confusing as it presented multiple, complex hypothetical infection sequences without clearly linking them to the epidemiological data.

To address this, we have deleted the confusing hypothetical sequences and replaced them with a detailed, evidence-based analysis of the reported microbiological differences:

The study by Newman et al. involved a relatively small series involving children aged 8 months to 9 years, with only five of six subjects undergoing culture analysis; four were positive, and three of these yielded Staphylococcus aureus. In contrast, the study by Uehara et al. was comparatively larger, analyzing 93 culture samples and including a population in which 36% were under 20 years old, indicating a relatively higher proportion of adults. Therefore, the differences in etiology between the two reports may be largely influenced by both the research scale and the age distribution of the patient populations.

Comment 12: Citations need to be uniformized according to the journals guideline

Response 12: Thank you for pointing out the lack of uniformity in our reference list. We have carefully reviewed the referencing guidelines for Reports (MDPI) and have made comprehensive revisions to ensure all citations adhere to the journal's standard format.

Ultimately, we would like to express our sincere gratitude to the editors and reviewers for their positive and constructive criticism. The manuscript has vastly benefited from your valuable and insightful comments and suggestions. We look forward to hearing from you and would be happy to address any further concerns, if required. We hope this further pushes the manuscript closer to publication in your esteemed journal.

All authors have read and approved the revised manuscript. 

Sincerely,

Koji Tajima, MD

Round 2

Reviewer 1 Report

Comments and Suggestions for Authors

The revised manuscript provides important laboratory information regarding the infecting agents.

Author Response

Dear Reviewer 1,

We are grateful for the thorough and constructive feedback provided on our revised manuscript. We have addressed all the comments and suggestions meticulously. We believe the manuscript has been significantly improved in terms of clarity, accuracy, and scientific rigor, particularly concerning the detailed microbiological analysis.

Our point-by-point responses are provided below, with the changes indicated using colored text in the revised manuscript (line numbers correspond to the clean version of the revised manuscript).

Comment 1: The revised manuscript provides important laboratory information regarding the infecting agents.

Response 1: We are very grateful for this positive feedback. We are pleased that the reviewer found the newly incorporated laboratory information to be important. The manuscript has been significantly strengthened by the reviewers' insightful guidance, leading to clearer and more rigorous presentation of the microbiological findings.

Ultimately, we would like to express our sincere gratitude to the editors and reviewers for their positive and constructive criticism. The manuscript has vastly benefited from your valuable and insightful comments and suggestions. We look forward to hearing from you and would be happy to address any further concerns, if required. We hope this further pushes the manuscript closer to publication in your esteemed journal.

All authors have read and approved the revised manuscript. 

Sincerely,

Koji Tajima, MD

Reviewer 3 Report

Comments and Suggestions for Authors

The article improved significantly but there are still some issues that needs to be addressed:

  • Based on the Figure 3 how was the sterility of the sampling technique assured? The umbilicus is not a sterile style and based solely on the imagine one could interpret that the pus was already there and therefore got contaminated with local flora thus the bacteria identified might not be significant for the infection.
  • Figure 7 – Since the stain and the image is not great, I would encourage you to erase the arrows and just present the image in a general manner.
  • Table 2 – Corynebacterium spp is just not enough. Diagnosing a bacterium based solely on Gram staining is not enough. I strongly encourage you to remove that part.
  • Regarding antimicrobial susceptibility testing, please extend the explanations because some readers might not use CLSI or might not have access to it.
  • Based on what was the grading of the bacteria 1+, 2+ etc. was performed.

Author Response

Dear Reviewer 3,

We are grateful for the thorough and constructive feedback provided on our revised manuscript. We have addressed all the comments and suggestions meticulously. We believe the manuscript has been significantly improved in terms of clarity, accuracy, and scientific rigor, particularly concerning the detailed microbiological analysis.

Our point-by-point responses are provided below, with the changes indicated using colored text in the revised manuscript (line numbers correspond to the clean version of the revised manuscript).

Comments 1: The article improved significantly but there are still some issues that needs to be addressed:

  • Based on the Figure 3 how was the sterility of the sampling technique assured? The umbilicus is not a sterile style and based solely on the imagine one could interpret that the pus was already there and therefore got contaminated with local flora thus the bacteria identified might not be significant for the infection.

Response 1: We thank the reviewer for this valuable comment. We fully agree with the reviewer that the umbilicus is a non-sterile site and that the potential for contamination by local flora is a critical factor for interpreting the initial sample.

We apologize if our previous response was ambiguous. We did not intend to suggest that the umbilical area itself was sterile.

Our primary concern during sampling was minimizing contamination that could arise during the collection procedure itself. Therefore, we ensured that the following steps were taken:

Aseptic technique: The swab for the first sample was handled using a strict sterile technique, to prevent contamination from hands, instruments, or non-sterile surfaces.

Reliable second sample: The second specimen was obtained by direct aspiration of the pus into a sterile syringe during the drainage procedure. This specific method is standard practice for culturing abscess contents, as it is designed to bypass surface flora and provide the specimen with the lowest possible risk of external contamination, thus supporting the clinical significance of the organisms detected.

We believe that the combination of our aseptic sampling technique and detection of multiple anaerobes from the lower-risk aspiration sample strongly supports their role as genuine pathogens in this closed-space infection, rather than as simple contaminants.

Comments 2: Figure 7 – Since the stain and the image is not great, I would encourage you to erase the arrows and just present the image in a general manner.

Response 2: We agree with the reviewer’s assessment regarding the limitations of the image quality and the difficulty in accurately identifying the arrow-pointed elements. 

As requested, we have removed all arrows from Figure 7 in the revised manuscript. 

We have also revised the legend for Figure 7 to be general yet informative, eliminating the specific mentions of the arrows, as follows: 

“Gram stain image (magnification: ×5000). A multi-bacterial pattern is observed, with gram-positive cocci, gram-positive rods, and gram-negative rods.” (Lines 107–108)

Comments 3: Table 2 – Corynebacterium spp is just not enough. Diagnosing a bacterium based solely on Gram staining is not enough. I strongly encourage you to remove that part.

Response 3: We are highly grateful for this crucial and definitive guidance. We fully agree with the reviewer that the presumptive identification of Corynebacterium sp. based solely on Gram staining is insufficient to conclusively establish its pathogenicity in the context of this report, especially given its nature as a common skin commensal. As the inclusion of this organism could weaken the overall scientific rigor of our findings, we have followed the reviewer's pertinent recommendation and removed all data and discussion related to Corynebacterium sp. from the manuscript to focus on the microorganisms with definitive identification. 

  • The reported number of isolated organisms is now four (four strict anaerobes). 
  • Table 2 has been revised to remove the column for Corynebacterium sp. 
  • The Abstract (lines 15 and 22), Case Presentation (Results), and Discussion sections (lines 157 and 163) have been comprehensively revised to reflect the final microbiological finding of four anaerobic pathogens.

Comments 4: Regarding antimicrobial susceptibility testing, please extend the explanations because some readers might not use CLSI or might not have access to it.

Response 4: We thank the reviewer for this constructive comment. We agree that providing context for the Clinical and Laboratory Standards Institute (CLSI) guidelines is essential to ensure clarity and accessibility for all readers, particularly those who utilize alternative standards (such as EUCAST) or have limited access to the full documents.

We have incorporated a concise explanation of the CLSI system's core principles and interpretation categories into the manuscript (in the Results section, following the mention of AST, lines 137–142).

The added text clarifies that the CLSI system:

  • Is a global standard for susceptibility testing.
  • Defines specific interpretive criteria (breakpoints) for microbial agents.
  • Categorizes results into the three clinical outcomes: susceptible (S), intermediate (I), and resistant (R), based on measured inhibition zone diameters. The purpose of these categories is to guide clinicians in selecting effective treatment.

We believe this addition fully addresses the reviewer's concern by providing sufficient context for the interpretation of our AST data without making the text overly dense.

Comments 5: Based on what was the grading of the bacteria 1+, 2+ etc. was performed.

Response 5: We thank the reviewer for this important question. We have consulted the specific criteria used by the microbiology laboratory for semi-quantitative grading of colony growth.  We have incorporated the precise definitions of the grading system into the manuscript to ensure data transparency (lines 121–126): 

  • 1+:A few colonies are observed across the entire field of view (estimated 103 cfu/mL). 
  • 2+:One to a few colonies are observed in every field of view (estimated 104 cfu/mL). 
  • 3+:Several tens or more colonies are observed in every field of view (estimated 105 cfu/mL or more).

Ultimately, we would like to express our sincere gratitude to the editors and reviewers for their positive and constructive criticism. The manuscript has vastly benefited from your valuable and insightful comments and suggestions. We look forward to hearing from you and would be happy to address any further concerns, if required. We hope this further pushes the manuscript closer to publication in your esteemed journal.

All authors have read and approved the revised manuscript. 

Sincerely,

Koji Tajima, MD